# What Motivates Patients with COPD to Be Physically Active? A Cross-Sectional Study

**DOI:** 10.3390/jcm10235631

**Published:** 2021-11-29

**Authors:** Sara Pimenta, Cândida G. Silva, Sofia Flora, Nádia Hipólito, Chris Burtin, Ana Oliveira, Nuno Morais, Marcelo Brites-Pereira, Bruno P. Carreira, Filipa Januário, Lília Andrade, Vitória Martins, Fátima Rodrigues, Dina Brooks, Alda Marques, Joana Cruz

**Affiliations:** 1Center for Innovative Care and Health Technology (ciTechCare), Polytechnic of Leiria, 2410-541 Leiria, Portugal; 5160250@my.ipleiria.pt (S.P.); candida.silva@ipleiria.pt (C.G.S.); sofia.flora@ipleiria.pt (S.F.); nadia.hipolito@ipleiria.pt (N.H.); nuno.morais@ipleiria.pt (N.M.); marcelo.b.pereira@ipleiria.pt (M.B.-P.); bruno.p.carreira@gmail.com (B.P.C.); 2School of Health Sciences (ESSLei), Polytechnic of Leiria, 2411-901 Leiria, Portugal; 3Coimbra Chemistry Centre, Department of Chemistry, University of Coimbra, 3004-535 Coimbra, Portugal; 4REVAL—Rehabilitation Research Center, Faculty of Rehabilitation Sciences, Hasselt University, B-3590 Diepenbeek, Belgium; chris.burtin@uhasselt.be; 5BIOMED—Biomedical Research Institute, Hasselt University, B-3590 Diepenbeek, Belgium; 6School of Rehabilitation Science, McMaster University, Hamilton, ON L8S 1C7, Canada; alao@ua.pt (A.O.); brookd8@mcmaster.ca (D.B.); 7Respiratory Medicine, West Park Healthcare Centre, Toronto, ON M6M 2J5, Canada; 8Lab 3R—Respiratory Research and Rehabilitation Laboratory, School of Health Sciences (ESSUA), University of Aveiro, 3810-193 Aveiro, Portugal; amarques@ua.pt; 9Centre for Rapid and Sustainable Product Development (CDRSP), Polytechnic of Leiria, 2430-028 Leiria, Portugal; 10Unidade de Saúde Familiar Pedro e Inês, ACeS Oeste Norte, 2460-492 Alcobaça, Portugal; 11Physical Medicine and Rehabilitation Department, Leiria Hospital Center, 2410-197 Leiria, Portugal; filipa.januario@chleiria.min-saude.pt; 12Pulmonology Department, Baixo Vouga Hospital Center, 3810-501 Aveiro, Portugal; m.lilia.andrade@hotmail.com; 13Pulmonology Department, District Hospital of Figueira da Foz, 3094-001 Figueira da Foz, Portugal; vitoria.b.martins@gmail.com; 14Pulmonology Department, Northern Lisbon University Hospital Centre (CHULN), 1769-001 Lisboa, Portugal; fatima.pneumo@gmail.com; 15Environmental Health Behaviour Lab, Faculty of Medicine, University of Lisbon (ISAMB/FMUL), 1649-028 Lisboa, Portugal; 16Institute of Biomedicine (iBiMED), University of Aveiro, 3810-193 Aveiro, Portugal

**Keywords:** active lifestyle, behaviour change, chronic obstructive pulmonary disease, motivation, physical activity, pulmonary rehabilitation

## Abstract

Motivation can be broadly defined as what moves people to act. Low motivation is a frequently reported factor for the reduced physical activity (PA) levels observed in patients with chronic obstructive pulmonary disease (COPD). This study assessed patients’ motives to be physically active, according to three pulmonary rehabilitation (PR) participation groups (Never PR, Previous PR and Current PR) and explored whether these motives were related to the PA levels and clinical characteristics. The motives to be physically active were assessed with the Exercise Motivation Inventory-2 (EMI-2, 14 motivational factors, five dimensions) and PA with accelerometry (PA groups: <5000 steps/day vs. ≥5000 steps/day). The clinical variables included symptoms, impact of the disease, exercise capacity and comorbidities. Ninety-two patients (67.4 ± 8.1 years, 82.6% male, forced expiratory volume in 1s (FEV_1_) 48.3 ± 18.9% predicted; 30.4% Never PR, 51% Previous PR and 18.5% Current PR) participated. The motivational dimensions related to health/fitness presented the highest scores (3.8 ± 1.1; 3.4 ± 1.3). The motives to be active were not significantly different between PA groups (*p* > 0.05) but having less symptoms and ≥two comorbidities were associated with higher scores in psychological/health and body-related motives, respectively (*p* < 0.05). The findings may encourage health professionals to actively explore with patients their motives to be physically active to individualise PA promotion.

## 1. Introduction

Optimising physical activity (PA) in patients with chronic obstructive pulmonary disease (COPD) is a major goal, as this population is highly inactive in daily life, and low physical activity (PA) levels are related to poor health outcomes (e.g., acute exacerbations, increased risk of hospitalisations and death) [1].

Low (or lack of) motivation is one of the most frequently reported factors for reduced PA levels in patients with COPD [2,3], in addition to the impact of symptoms such as dyspnoea on exertion and fatigue [4,5]. Motivation can be broadly defined as what moves people to act [6]. The Self-Determination Theory (SDT) provides a framework to understand motivation for behaviour change, and it is one of the most recommended theories to be used in PA promotion [7]. According to this theory, human motivation is a dynamic process that oscillates in a continuum from amotivation and extrinsic motivation to intrinsic motivation [8,9]. Intrinsic motivation refers to the individual’s motivation to perform PA because it is inherently interesting or enjoyable (i.e., more autonomous motivation), while extrinsic motivation refers to motivation that produces specific outcomes in terms of rewards or avoided punishments [6]. Even if extrinsic, motives can be more autonomous (e.g., feeling more energetic and avoiding health problems) or less autonomous (e.g., following someone’s advice). Although an individual can be both extrinsically and intrinsically motivated at the same time [10], the more autonomous/self-determined the motives are, the more likely an individual is to achieve and maintain that behaviour [9]. Therefore, understanding the individual motives for engaging in PA may help in the development of effective interventions to improve patients’ PA levels.

Pulmonary rehabilitation (PR) is the cornerstone of nonpharmacological treatment in COPD, with well-documented benefits on symptoms, exercise tolerance and health-related quality of life [11]. It consists of a comprehensive intervention that includes (but is not limited to) exercise training, education and behaviour changes to promote better health and long-term adherence to health-enhancing behaviours. Motivation is not a prerequisite for enrolling in PR; rather, it is one of its major goals [11]. Hence, it would seem the ideal intervention to promote PA behaviours, since it arguably contains the necessary ingredients to achieve significant and lasting increases in PA in patients with COPD [12]. However, there is still inconsistent evidence of the impact of PR on patients’ PA levels [13]. Only one study found that more autonomous forms of motivation were significantly related to better exercise behaviours following PR in patients with COPD [14]. However, the study focused only on a specific type of PA (exercise) and was limited to patients who participated in PR. Moreover, patients’ motives to be more physically active were not investigated, although they can help tailoring interventions for PA promotion [10]. This study aimed to assess motives to be physically active in patients with COPD, exploring differences in motives among patients with no, previous and current PR exposure, and to investigate whether these motives were related to patients’ PA levels measured objectively. We hypothesised that motives to be physically active would be different (more autonomous) in patients with previous/current exposure to PR compared to no PR exposure and in active patients compared to those inactive. As a secondary aim of the study, we explored the relationship between motives to be physically active and patients’ clinical characteristics (i.e., symptoms, impact of the disease, exercise capacity and comorbidities).

## 2. Materials and Methods

### 2.1. Study Design

This cross-sectional study was part of a larger study (OnTRACK, ref. POCI-01-0145-FEDER-028446) and was reported according to the STrengthening the Reporting of OBservational studies in Epidemiology (STROBE) statement [15]. No sample size calculation was performed, since this was a secondary and exploratory study, and there were no reference studies on the topic that could be used. Ethical approval was obtained prior to study commencement (more information in the Institutional Review Board Statement section). Written informed consent was obtained from all participants prior to data collection. Data were collected between January 2019 and March 2020.

### 2.2. Participants

Patients with COPD were recruited from the health units collaborating in the study: Hospital Centres of Leiria, Baixo Vouga and Universitário Lisboa Norte, the District Hospital of Figueira da Foz, the Santiago Family Health Unit of the Primary Care Centre Dr Arnaldo Sampaio and the Respiratory Research and Rehabilitation Laboratory of the School of Health Sciences, University of Aveiro (Lab3R-ESSUA). Patients were identified by their physicians after confirming the fulfilment of the eligibility criteria and interest to participate.

To be eligible, patients had to be: (1) diagnosed with COPD according to the Global Initiative for Chronic Obstructive Lung Disease (GOLD) criteria [16], (2) clinically stable (i.e., without hospital admissions or exacerbations in the previous month) and (3) able to provide written informed consent. Exclusion criteria consisted of: (1) another clinical condition that could limit PA (e.g., significant neurological or musculoskeletal disorders), (2) another respiratory disease as the primary diagnosis, (3) unstable and/or severe cardiovascular disease, (4) a recent history of neoplasia or immune disease and (5) signs of cognitive impairment or other conditions that could preclude patients from understanding and/or cooperating in the study.

### 2.3. Data Collection

Eligible patients were contacted by telephone by the researchers who provided detailed information about the study. Those who agreed to participate were invited to an appointment to sign the informed consent form and collect the data. Data collection was carried out in the health unit where the patient was recruited or at the facilities of the Center for Innovative Care and Health Technology (ciTechCare) and Lab3R-ESSUA, according to the availability of participants and health services.

The sociodemographic (age and sex), anthropometric (height and weight to compute the body mass index—BMI) and clinical data (as described below) were first collected. Lung function was assessed through spirometry using a portable spirometer (Microloop, Carefusion, Kent, UK), according to the guidelines of the European Respiratory Society (ERS) [17] for further classification of the severity of the airflow limitation (GOLD 1–4) [16].

Information regarding PR participation was also collected for the further classification of patients into three groups: (1) patients who never engaged in a PR programme (Never PR), (2) patients who participated in a PR programme before (Previous PR) and (3) patients who were attending a PR programme at the moment of data collection (Current PR). PR programmes had to include at least one component of exercise training and one component of education and psychosocial support during at least 8 weeks, following the guidelines [11]. Information on the time since the last PR (for the Previous PR group) and number of sessions attended (for the Current PR group) was also collected to characterise the sample.

#### 2.3.1. Motives to Be Physically Active

The Portuguese version [18] of the EMI-2 [10] was used to assess patients’ motives to be physically active. This is a self-administered tool that enables the identification of the main motivational factors for exercise reported by an individual, and it is applicable to both exercisers and non-exercisers [10]. It contains 51 items on a 6-point Likert scale (0 = “nothing true for me” to 5 “completely true for me”) that assess 14 motivational factors that can be grouped into 5 dimensions (Table 1). Scores are obtained for each motivational factor by calculating the average of the items corresponding to that factor. For each dimension, the average of the factors pertaining to that dimension is calculated. Higher values reflect a greater weight of this motivational factor/dimension for exercise/PA.

The EMI-2 has been applied in a variety of studies and samples [19,20,21,22,23,24], including older adults [19,23]. It was also found to discriminate between individuals at different stages of behaviour changes [25], which may be valuable for future interventions [22]. The EMI-2 showed good internal consistency in the Portuguese validation studies (Cronbach’s α ≥ 0.70) [18,26] and in the present study (0.721 ≤ α ≤ 0.905, except for Health Pressure α = 0.559; Appendix A).

The motivational factors of the EMI-2 are related to the conceptual basis of the SDT [8,9]. Table 1 presents the relationship between the motivational factors and the motivation types identified by the SDT (i.e., intrinsic, extrinsic, other in the legend of the table) based on previous research [20,21,22,24], although it has been argued that it is difficult to categorise some motives as being exclusively of one type [10].

#### 2.3.2. Physical Activity

Daily PA was assessed using an accelerometer—Actigraph GT3X+ (Actigraph Pensacola, Pensacola, FL, USA), which has been validated for the COPD population [27,28]. Participants were instructed to use the device for 7 consecutive days from the moment they woke up until they went to bed (except for bathing or swimming) and were informed about their correct positioning, i.e., at the waist on the dominant side with an elastic belt.

Data were extracted from the accelerometer using Actilife software v6.10.4 (Actigraph Pensacola, Pensacola, FL, USA). The PA assessment was judged adequate and representative if it included at least 4 valid days (i.e., days with at least 8 hours of wear time) [29]. All valid days were included in the analyses. Daily PA included the number of steps per day and time spent in moderate-to-vigorous PA (MVPA) (in min/day). The cut-off points for MVPA intensity were defined according to Freedson et al. [30] (1952 ∞ counts per minute). The number of daily steps was also analysed using the step-defined sedentary lifestyle index proposed by Tudor-Locke et al. [31]: <5000 steps/day (sedentary lifestyle); ≥5000 steps/day (active lifestyle).

#### 2.3.3. Clinical Variables

The severity of dyspnoea during activities was assessed using the mMRC scale (score range: 0–4, with 4 being the most dyspnoeic) [32]. An mMRC of ≥2 was considered a threshold for separating “less breathlessness” from “more breathlessness” [16].

The CAT is an 8-item unidimensional measure that evaluates the impact of COPD (score range: 0–40), with higher scores representing a higher symptom burden [33]. The cut-off point established for symptom severity is 10 [16].

Subjective fatigue was measured using the Portuguese version [34] of the CIS20-SF [35]. This is one of the subscales of the CIS20 instrument, which refers to the perception of fatigue in the past two weeks. It consists of 8 items that are scored on a 7-point Likert scale from 1 (“Yes, that is true”) to 7 (“No, that’s not true”). Scores range from 8 to 56, with higher scores indicating a higher degree of fatigue. The results can be categorised into normal fatigue (<27 points) and abnormal fatigue (≥27 points), which can be further divided into mild (27–35 points) and severe (≥36 points) fatigue [35,36]. CIS20-SF is a validated tool [35] that has been increasingly used in patients with COPD [37,38,39,40].

The exercise capacity was assessed with the 6-min walk test (6MWT), according to the ATS/ERS standards [41] on a 30-meter indoor walking track. The distance (6MWD) and percentage predicted [42] were reported. Participants were further categorised as having low (6MWD <350 m) or high (6MWD ≥350 m) exercise capacity [43,44].

The number of comorbidities was assessed based on self-reporting using the COPD Comorbidity Checklist [45] and dichotomised into two categories: <2 comorbidities; ≥2 comorbidities.

### 2.4. Data Analysis

Statistical analyses were conducted using the IBM Statistical Package for Social Sciences (SPSS) software v27 (IBM, Armonk, New York, USA), with the significance level set at 0.05. Graphs were created using package ggplot2 [46] under R software version 4.0.2 (The R Foundation, Vienna, Austria) [47].

Descriptive statistics were used to describe the sample characteristics and the results of the EMI-2. Data were presented as frequencies, mean ± standard deviation (SD) or median (interquartile range, IQR), as appropriate.

Normality of the data was assessed using the Kolmogorov–Smirnov test. ANOVA and independent *t*-tests (or their equivalent nonparametric tests, depending on the (non-)normality of the data distribution) and chi-square tests (for categorical variables) were used to compare sample characteristics among the three PR groups. The motives to be physically active (EMI-2) were compared using the same tests, considering the PR participation (main aim), PA categories (sedentary vs. active lifestyle) and categories regarding the clinical characteristics (GOLD 1–4 and ABCD; mMRC, CAT and CIS20-SF cut-off scores; 6MWD and number of comorbidities). When three or more groups were compared and a significant result was obtained, post hoc pairwise comparisons were performed to identify the different groups (ANOVA: Scheffe test; Kruskal–Wallis test: pairwise comparisons with Bonferroni correction for multiple testing).

In case of missing data, the number of valid cases for each variable was identified and provided in the Appendix A. In the main test, the maximum number of valid cases is presented.

## 3. Results

### 3.1. Participants

A total of 132 patients were identified. From these, 18 refused to participate, 7 were not available at the time of data collection and 1 died before data collection. Additionally, two withdrew from participating, two reported having had an exacerbation in the previous month and two had an exacerbation during the data collection period (accelerometry). Eight patients did not complete the EMI-2 and, therefore, were excluded from the analysis.

The final sample was composed of 92 participants, with a mean (±SD) age of 67.4 ± 8.1 years, a FEV_1_ of 48.3 ± 18.9% predicted and a BMI of 26.2 ± 4.7 kg/m^2^. Most participants were male (82.6%) and former smokers (69.6%) and had two or more comorbidities (81.5%) (Table 2). The most reported comorbidities were hypertension (*n* = 37, 40.7%), dyslipidaemia (*n* = 31, 33.7%), anxiety (*n* = 16, 17.4%) and diabetes mellitus (*n* = 13, 14.1%). The participants’ characteristics are presented in Table 2.

Regarding PR participation, 28 patients had never attended PR (Never PR), 47 had participated in a PR programme before (Previous PR; median (IQR) of 6 (3–11.5) months since PR) and 17 were attending a PR programme in the moment of data collection (Current PR, 4 (4.5–7.8) sessions attended) (Table 2). The groups were significantly different in FEV_1_% predicted and mMRC (*p* < 0.05), with the “Never PR” showing better results than the “Previous PR” and the “Current PR”. A significant difference was also found in the CAT scores (*p* = 0.045); however, when applying a more conservative method to assess pairwise comparisons, there was only a trend for differences between the “Never PR” and “Current PR” (*p* = 0.066).

### 3.2. Motives to Be Physically Active and PR Participation

Overall, the patients’ motives to be physically active were mostly related to health (3.8 ± 1.1) and fitness (3.4 ± 1.3) dimensions, followed by psychological dimensions (score 2.9 ± 1.3) (Appendix A). The Ill Health Avoidance and Positive Health were the motivational factors scoring the highest (3.9 ± 1.2 and 4.0 ± 1.2, respectively). When considering the PR groups, significant differences were found for the factor Enjoyment (*p* = 0.047, Figure 1). Post hoc pairwise comparisons showed a near significant difference between the “Never PR” and “Previous PR” groups (2.4 ± 1.7 vs. 3.3 ± 1.3, *p* = 0.055). No additional differences were found (*p* > 0.05).

### 3.3. Motives to Be Physically Active and Physical Activity

No significant differences were found in the motivational dimensions and factors to be physically active between the sedentary and active lifestyle groups (*p* > 0.05; Appendix A).

### 3.4. Motives to Be Physically Active and Clinical Characteristics

Significant differences were found in the motives to be physically active between patients in different symptom categories (Figure 2 and Appendix A). Specifically, patients with less symptoms (i.e., CAT < 10, mMRC < 2 and CIS20-SF < 27) presented significantly higher scores in the factors Revitalisation and Positive Health than those with more symptoms (*p* < 0.05). Patients with a lower score in the CAT and the CIS20-SF also showed significantly higher scores in Enjoyment (CAT: 3.5 ± 1.0 vs. 2.7 ± 1.6; CIS20-SF: 3.7 ± 1.1 vs. 2.7 ± 1.5; *p* < 0.05) and Ill Health Avoidance (CAT: 4.4 ± 0.8 vs. 3.8 ± 1.3; CIS20-SF: 4.3 ± 1.1 vs. 3.8 ± 1.2; *p* < 0.05). Affiliation was only significantly different between the CAT groups (CAT <10: 2.6 ± 1.2 vs. CAT ≥10 1.9 ± 1.6, *p* = 0.021).

Patients with two or more comorbidities presented higher scores in Body-Related Motives than those with one or none (0–1: 1.3 ± 1.2 vs. ≥2: 2.2 ± 1.4, *p* = 0.013), specifically Weight management (*p* = 0.005) and Appearance (*p* = 0.031) (Appendix A).

No significant differences were found for the GOLD 1–4, GOLD ABCD and 6MWD groups (Appendix A).

## 4. Discussion

This study explored the motives to be physically active in patients with COPD considering PR participation, PA levels and clinical characteristics. The results showed that patients highly valued health and fitness motives, and there was a trend for patients who participated in PR to identify enjoyment in being physically active when compared to those who never enrolled in PR. The motives to be physically active were not different between patients with sedentary or active lifestyles (i.e., <5000 vs. ≥5000 steps/day), but having less symptoms and two or more comorbidities was associated with higher motivational levels related to psychological/health and body-related motives, respectively. The findings may encourage health professionals to actively explore with patients their motives to be physically active to individualise PA promotion.

Overall, the health and fitness motives to be physically active were most highly valued by patients. This finding may indicate patients’ recognition of the important role of PA on the maintenance of health and functional capacity. One previous study found that exercise motivation (assessed using the EMI-2) varies significantly as a function of age and that health and fitness factors elicited higher motivation responses in young, old adults (50–64 years) and old adults (65+ years) than in younger people (25–34 years) [23]. Therefore, the findings from the present study may be related to the normal changes in motives to be physically active across the life span. We did not analyse age-related differences in motives due to the small variation in the age range of participants (43–81 years).

Patients who participated in PR before (Previous PR) seemed to find PA more enjoyable than those who never enrolled in PR, although the results were only nearly significant. Enjoyment has been considered an intrinsic motive to be physically active and is related to the positive experience/feeling of exercising [19]. Although exercise training during PR involves mostly less autonomous (extrinsic) forms of motivation and the goals are mainly set to attain disease-related outcomes, the PR environment is likely to increase autonomous motivations by supporting individuals’ basic psychological needs, according to the SDT: autonomy, competence and relatedness [48]. Therefore, PR participation may influence patients’ perception of PA positively and help changing their attitudes towards PA (although not necessarily their behaviour), as previously suggested [49]. Scheermesser et al. [49] assessed patients’ personal experience with PA after a 12-week PR programme and found that patients’ PA behaviour might be influenced by their concept of PA (i.e., what PA means to them) and the quality of motivation, with more intrinsically motivated patients reporting higher PA levels. The findings from that study also indicated that those individuals who participated in the PR programme combined with embedded counselling tended to be more active and intrinsically motivated than the ones participating in PR only, suggesting that counselling should be included in PR programmes to support patients’ PA behaviours [49].

The absence of differences in Enjoyment between patients who never enrolled in PR and patients currently attending PR may be related to that fact that, although research has consistently shown that intrinsic motives play a major role in maintaining an active behaviour [9], the intrinsic rewards of exercising may not be immediately apparent to those beginning an exercise/PR programme. The initial adoption of exercise/PA is more likely to be motivated by perceptions of health and fitness benefits [22].

There was no significant association between motives to be physically active and PA behaviour. These results differ from a previous study conducted in patients with COPD aimed to examine reasons for not walking and their relationship with directly measured step counts [50]. The authors found that patients who were most motivated to walk each day reached 1093 more daily steps than those who were less motivated (adjusting for age and FEV_1_% predicted). Although the patients identified several reasons to walk as “very important”, only reasons related to preventing future health problems (health reasons) and feeling less tension and stress (stress management—a more intrinsic reason) were significantly related to the daily step count [50]. Nevertheless, findings need to be interpreted with caution, since the questionnaire used to assess the motivations and reasons for not walking was developed by the authors based on face validity from clinical experience, without further validation. The relationship between motives to be physically active and exercise/PA behaviour was also investigated in a systematic review including studies with a range of samples and settings (although none conducted on patients with COPD) [9]. The authors concluded that more intrinsic motives (e.g., enjoyment, affiliation and challenge) were positively associated with PA/exercise participation, while findings for fitness/health and body-related motives yielded mixed results [9]. Our findings are not in line with the results of this review. Further research is warranted to explore the relationship between motives to be physically active and PA behaviour in patients with COPD, as well as the potential moderating factors of this relationship.

Patients with less symptom burdens presented significantly higher motivational levels in the factors related to health benefits (Positive Health and Ill Health Avoidance) and personal intrinsic needs (Revitalisation, Enjoyment and Affiliation) than those with more symptoms, while no differences were found in the groups with different exercise capacity levels and disease severity/impact. These findings suggest that symptoms (but not exercise capacity or disease severity/impact) have a role in patients’ motivation to be physically active. In fact, the most significant self-reported barrier towards PA was found to be low motivation [2], and it was associated with a fear of breathlessness, which supports our findings. It can therefore be hypothesised that more symptomatic patients feel less autonomously motivated to be physically active and reduce their PA as a way of avoiding an increase in symptoms. In the present study, patients with better scores in the mMRC and CIS20-SF categories (i.e., mMRC < 2; CIS20-SF < 27) presented better PA levels when compared to those with higher symptom burdens (*p* < 0.05, data not shown), thus supporting our hypothesis. Revitalisation, Enjoyment and Affiliation are considered intrinsic factors, i.e., are concerned with experiences of competence and interest in the activity per se [10], while health factors cannot be easily defined as intrinsic or extrinsic factors, as it depends on what the motive means to the individual (e.g., general health promotion, feeling more energy/vitality and avoiding health problems) [9,10]. Even if extrinsic, these factors may be important to some individuals, particularly in the decision to adopt PA in the first place [10].

Having two or more comorbidities was associated with higher scores in body-related motives to be physically active, specifically weight management and appearance. Al- though these motives are often associated with a desire to attain ego enhancement, social approval or feelings of self-worth (less autonomous motives) [19], they may also be related to the fact that 58.7% of the patients in the ≥two comorbidities group reported having a metabolic disease (e.g., dyslipidaemia or diabetes—data not shown), and thus, they may be more aware of the need to improve their body composition. This topic deserves further investigation.

### 4.1. Implications for Pulmonary Rehabilitation and Future Research

The findings from the present study do not indicate one clear path forward, but they do signal that the optimisation of PA interventions may include the identification of the individuals’ motivations to be physically active considering their clinical characteristics (particularly their symptoms). These interventions may require a comprehensive approach, including PR and PA counselling, to address both the physiological and behavioural factors influencing a PA behaviour.

It should be noted that different motives naturally coexist in the same individual, and the balance between intrinsic and extrinsic motives may determine the achievement of the desirable outcomes [9]. Therefore, being able to identify individuals’ dominant motives to be physically active can give important insight into their overall motivation and PA behaviours. Generally, more self-determined/autonomous (intrinsic) motives are associated with sustained engagement in the behaviour. Still, PA promotion interventions should recognise all motives to be physically active, even if less autonomous (extrinsic), as they can be relevant to initiating behaviours and can ultimately promote the shift to more autonomous motives [9].

It is also important to acknowledge that, as a complex behaviour, PA may be influenced by an interplay of individual, social and environmental factors, including (but not limited to) motivation, which may act as barriers/facilitators for PA behaviours [51]. Some of the factors influencing PA in patients with COPD have been identified in the literature [3,52] and may be different from one patient to another. Therefore, there is a need for an individual approach to understand the factors underlying patients’ decisions to be (or not) physically active in order to develop comprehensive and individualised PA promotion interventions.

### 4.2. Strengths and Limitations of the Study

To the authors’ best knowledge, this is the first study assessing the motives for being physically active in patients with COPD using a validated scale and exploring their relationship with PR participation, PA levels and clinical characteristics. Data were collected in six health units (four hospitals, one primary health care centre and one community centre) and included objectively measured PA behaviours, which are strengths of the study. Nevertheless, there were some limitations that need to be acknowledged. The first was related to the fact that EMI-2 assesses the motives for exercising instead of the daily PA in general. Although this scale was developed and validated to be used in both exercisers and non-exercisers [10], the questions are mostly related to exercise (i.e., a structured form of PA [53]). Additionally, EMI-2 has not been specifically validated for patients with COPD, although it has been used in adult (and older adult) populations [19,23]. Nevertheless, the internal consistency results of the EMI-2 obtained in the present study were good and above the minimum recommended [54]. Additionally, sociodemographic and socioeconomic characteristics and culture were not addressed in this study, although they may influence exercise motivation [19,22,23,55]. Furthermore, we did not collect information on the number of PR sessions completed in the “Previous PR” group and on the specific types of PA patients were doing during data collection (e.g., sports, exercise and other PA), although it may be important to have a deeper understanding of the relationships between motives and PA [9]. Patients in the Never PR group had better lung function than those in the other PR groups, which may limit the generalisability of results. This finding was somewhat expected, as, in clinical practice, the severity of lung function impairment is often (although inadequately) used as a criterion for patient referral to and enrolment in PR [56]. Nevertheless, when the motives to be physically active were compared among the GOLD 1–4 grades, no significant differences were found, indicating that airflow obstruction may not play a role in the motives to be physically active. Finally, the sample size was relatively small, and there were some missing values and unbalanced groups that may have impacted the results (Appendix A).

## 5. Conclusions

The motives to be physically active in patients with COPD were mostly related to health and fitness, and there was a trend for patients who participated in PR to perceive PA as more enjoyable than those who never enrolled in PR. Nevertheless, the motives to be physically active were not associated with PA behaviours. The findings suggest that symptom burdens and comorbidities may influence patients’ motives to be physically active. More research is needed to support these findings.

## Figures and Tables

**Figure 1 jcm-10-05631-f001:**
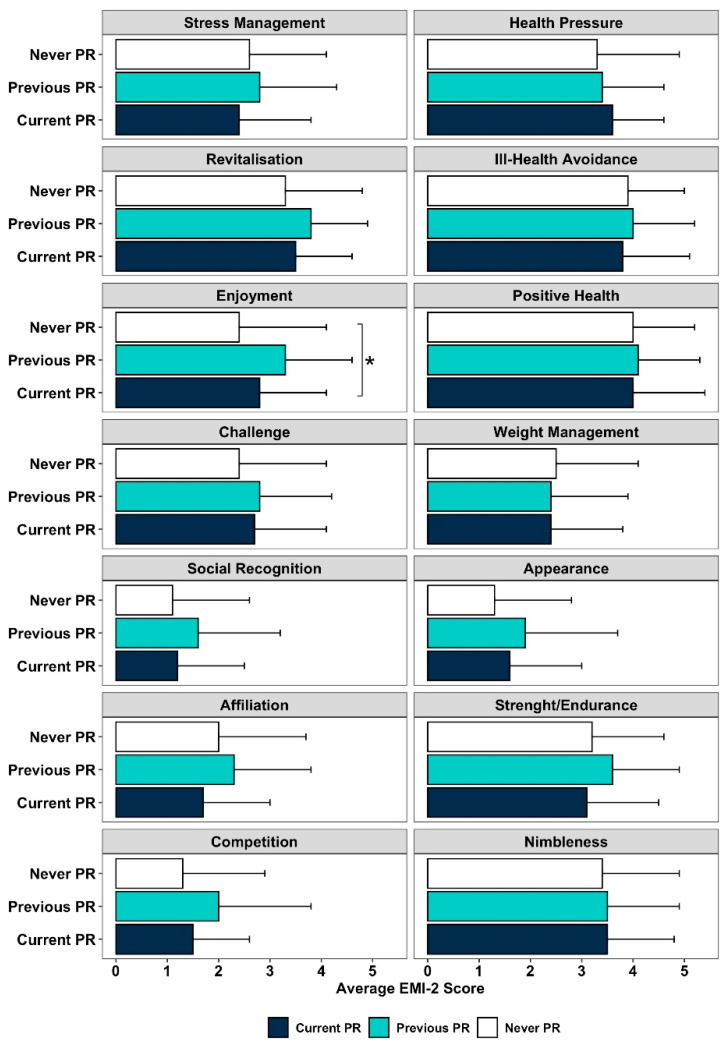
Motivational factors of the Exercise Motivation Inventory-2 (EMI-2) among the PR participation groups (*n* = 92; * *p* < 0.05).

**Figure 2 jcm-10-05631-f002:**
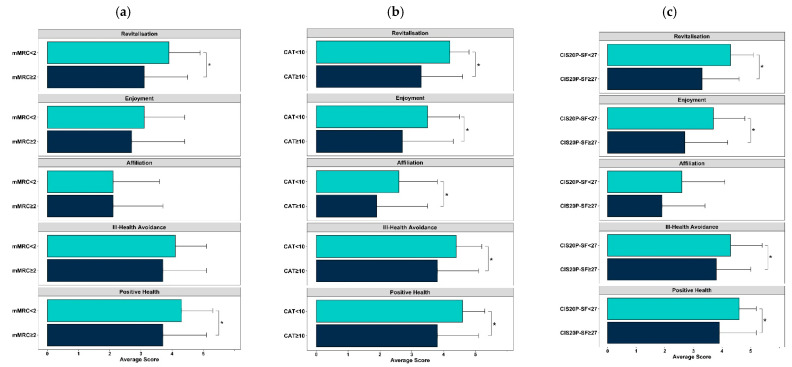
Motivational factors of the Exercise Motivation Inventory-2 (EMI-2) among the symptoms using the COPD Assessment Test (CAT), modified Medical Research Council Dyspnoea Scale (mMRC) and Checklist of Individual Strength—Subjective Fatigue (CIS20-SF) (* *p* < 0.05). (**a**) mMRC. (**b**) CAT. (**c**) CIS20-SF.

**Table 1 jcm-10-05631-t001:** Motivational dimensions and factors of the Exercise Inventory Motivation 2 (EMI-2).

Dimensions	Factors	Items
Psychological Motives	Stress Management ^a^	6–20–34–46
	Revitalisation ^a^	3–17–31
	Enjoyment ^a^	9–23–37–48
	Challenge ^a^	14–28–42–51
Interpersonal Motives	Social Recognition ^b^	5–19–33–45
	Affiliation ^a^	10–24–38–49
	Competition ^c^	12–26–40–50
Health Motives	Health Pressure ^b^	11–25–39
	Ill Health Avoidance ^b^	2–16–30
	Positive Health ^c^	7–21–35
Body-Related Motives	Weight Management ^b^	1–15–29–43
	Appearance ^b^	4–18–32–44
Fitness Motives	Strength/Endurance ^c^	8–22–36–47
	Nimbleness ^a^	13–27–41

^a^ Intrinsic motives, ^b^ Extrinsic motives and ^c^ Others.

**Table 2 jcm-10-05631-t002:** Participants’ sociodemographic and clinical characteristics (*n* = 92).

	Total Sample (*n* = 92)	Never PR (*n* = 28)	Previous PR (*n* = 47)	Current PR (*n* = 17)	*p*-Value
**Age, mean (SD)**	67.4 (8.1)	65.5 (8.7)	68.3 (6.4)	68.0 (11.2)	0.353 ^a^
**Sex, *n* (%)**					
Female	16 (17.4)	9 (32.1)	7 (14.9)	-	- ^d^
Male	76 (82.6)	19 (67.9)	40 (85.1)	17 (100.0)	
**Smoking status, *n* (%)**					
Current smoker	20 (21.7)	9 (32.1)	9 (19.1)	2 (11.8)	- ^d^
Former smoker	64 (69.6)	17 (60.7)	33 (70.2)	14 (82.4)	
Never smoker	8 (8.7)	2 (7.1)	5 (10.6)	1 (5.9)	
**BMI (kg/m^2^), mean (SD)**	26.2 (4.7)	26.9 (4.1)	25.1 (4.9)	28.0 (4.4)	0.055 ^c^
**FEV_1_ (% predicted)** **, mean (SD)**	48.3 (18.9)	61.4 (17.0)	43.3 (16.3)	41.2 (19.2)	<0.001 * ^a,e^
**GOLD 1** **–4, *n* (%)**					
GOLD 1	4 (4.4)	3 (11.1)	-	1 (5.9)	- ^d^
GOLD 2	35 (38.5)	17 (63.0)	14 (29.8)	4 (23.5)	
GOLD 3	36 (39.6)	6 (22.2)	24 (51.1)	6 (35.3)	
GOLD 4	16 (17.4)	1 (3.7)	9 (19.1)	6 (35.3)	
**GOLD ABCD, *n* (%)**					
A	40 (45.5)	16 (59.3)	19 (41.3)	5 (33.3)	0.718 ^b^
B	16 (18.2)	4 (14.8)	9 (19.6)	3 (20.0)	
C	15 (17.0)	4 (14.8)	8 (17.4)	3 (20.0)	
D	17 (19.3)	3 (11.1)	10 (21.7)	4 (26.7)	
**Comorbidities, *n* (%)**					
0 to 1	17 (18.5)	5 (17.9)	10 (21.3)	2 (11.8)	0.684
≥2	75 (81.5)	23 (82.1)	37 (78.7)	15 (88.2)	
**CAT, mean (SD)**	14.8 (8.6)	12.8 (11.1)	15.3 (7.7)	16.5 (6.0)	0.045 * ^c^
**mMRC, median (Q1** **to Q3)**	1 (1 to 2)	1 (1 to 2)	2 (1 to 2)	2 (1 to 2.5)	0.009 * ^c,e^
**CIS20-SF, *n* (%)**					
Normal fatigue	26 (29.5)	10 (38.5)	14 (31.1)	2 (11.8)	0.127 ^b^
Mild fatigue	25 (28.4)	5 (19.2)	16 (35.6)	4 (23.5)	
Severe fatigue	37 (42.0)	11 (42.3)	15 (33.3)	11 (64.7)	
**6MWD (m), mean (SD)**	421.8 (95.7)	446.8 (68.6)	414.0 (96.1)	400.8 (126.0)	0.223 ^a^
**6MWD% predicted, mean (SD)**	85.3 (19.1)	88.9 (11.7)	85.0 (20.9)	80.1 (23.1)	0.326 ^a^
**PA, mean (SD)**					
MVPA (min/day)	29.3 (25.1)	35.7 (32.7)	24.3 (19.2)	32.6 (24.2)	0.409 ^c^
No. Steps	5178.9 (3168.3)	6217.5 (4030.0)	4560.1 (2457.2)	5203.6 (3089.1)	0.338 ^c^

Abbreviations: BMI, body mass index; CAT, COPD assessment test; CIS20-SF, Checklist of Individual Strength—Subjective Fatigue; FEV_1_, Forced expiratory volume in the 1st second; GOLD, Global Initiative for Chronic Obstructive Lung Disease; mMRC, Modified Medical Research Council Dyspnea Scale; MVPA, moderate-to-vigorous PA; PA, physical activity; SD, Standard Deviation; 6MWD, 6-min walking distance. ^a^ ANOVA Test; ^b^ chi-squared test; ^c^ Kruskal–Wallis test. ^d^ Chi-squared test was not possible to perform due to some cells having an expected count less than 5. ^e^ Differences between the “Never PR” and other two groups (“Previous PR” and “Current PR”). * *p* ≤ 0.05.

## Data Availability

The data presented in this study are available on request from the corresponding author. The data are not publicly available due to ethical issues.

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
