# Peer review of "What Motivates Patients with COPD to Be Physically Active? A Cross-Sectional Study"

_jcm, 2021, doi:10.3390/jcm10235631_

Round 1
Reviewer 1 Report
The authors aimed, as a primary outcome, to investigate COPD patients motives to be physically active using two methods: a self- reported questionnaire (the EMI-2) and an accelerometer (Actigraph GTX3+). The study has a cross-sectional design, comparing 3 different groups of patients: no previous PR, PR and previous PR. The study and study findings are interesting and can be applicable in the clinical work with COPD patients to better understand motives for being/not being physical active and provide with a proper advice.
I have following comments/suggestions to the authors:
- The authors state that the follow Strobe guidelines. But two issues are missing in the manuscript: a) What there any initial hypothesis? At least for the primary outcomes?
- Please provide a sample size calculation. The authors may not have sufficient power to detect a significant difference between groups. The authors mention the relative small sample and an unbalanced group as a study limitation. I suggest the authors to elaborate on the small sample issue by providing a sample size calculation for expected differences for the EMI-2/ objective measure of PA. The results for “motives to be physically active and PA” page 7, lines 256-259 may all in all be due to unpowered sample and must be addressed properly.
Furthermore:
Was there any reason for choosing to analyse the cohort into 3 separate groups (other than the small samples for non-PR and actual PR group)? E.g. change of motivation for being PA over time in the "previous PR group"? Other reasons? This is not clearly described under methods – or elaborated under discussion. A major challenge for patients with chronic diseases (including COPD) patients is to keep with appropriate PA levels over time.
Reviewer 2 Report
1. Gender differences were not specified in the results. It should be added to the results and revised the table that there is a gender difference between the groups.
2. Was the results adjusted for difference of the FEV1 value a? Authors should add this topic to the discussion as well. Could the fact that the Never PR group be better, affected the results? 3. The average number of sessions completed in pulmonary rehabilitation should be added.
4. When divided into groups (PA, GOLD, mMMRC, CAT, etc.), were the physical and demographic characteristics of the groups similar? If there is a difference, the authors should provide information on this issue.
